# Mineralized Polyvinyl Alcohol/Sodium Alginate Hydrogels Incorporating Cellulose Nanofibrils for Bone and Wound Healing

**DOI:** 10.3390/molecules27030697

**Published:** 2022-01-21

**Authors:** Ragab E. Abouzeid, Ahmed Salama, Esmail M. El-Fakharany, Vincenzo Guarino

**Affiliations:** 1Cellulose and Paper Department, National Research Centre, 33 El-Bohouth St., Dokki, Giza 12622, Egypt; ahmed_nigm87@yahoo.com; 2Protein Research Department, Genetic Engineering and Biotechnology Research Institute (GEBRI), City of Scientific Research and Technological Applications (SRTA-City), New Borg EL Arab, Alexandria 21934, Egypt; esmailelfakharany@yahoo.co.uk; 3Institute of Polymers, Composite and Biomaterials, National Research Council of Italy, Mostra D’Oltremare, Pad 20, V. J.F. Kennedy 54, 80125 Naples, Italy

**Keywords:** cellulose, calcium phosphates, TEMPO reaction, cytotoxicity, hybrid materials

## Abstract

Bio sustainable hydrogels including tunable morphological and/or chemical cues currently offer a valid strategy of designing innovative systems to enhance healing/regeneration processes of damaged tissue areas. In this work, TEMPO-oxidized cellulose nanofibrils (T-CNFs) were embedded in alginate (Alg) and polyvinyl alcohol (PVA) solution to form a stable mineralized hydrogel. A calcium chloride reaction was optimized to trigger a crosslinking reaction of polymer chains and mutually promote in situ mineralization of calcium phosphates. FTIR, XRD, SEM/EDAX, and TEM were assessed to investigate the morphological, chemical, and physical properties of different mineralized hybrid hydrogels, confirming differences in the deposited crystalline nanostructures, i.e., dicalcium phosphate dehydrate (DCPDH) and hydroxyapatite, respectively, as a function of applied pH conditions (i.e., pH 4 or 8). Moreover, in vitro tests, in the presence of HFB-4 and HSF skin cells, confirmed a low cytotoxicity of the mineralized hybrid hydrogels, and also highlighted a significant increase in cell viability via MTT tests, preferentially, for the low concentration, crosslinked Alg/PVA/calcium phosphate hybrid materials (<1 mg/mL) in the presence of hydroxyapatite. These preliminary results suggest a promising use of mineralized hybrid hydrogels based on Alg/PVA/T-CNFs for bone and wound healing applications.

## 1. Introduction

In recent years, sustainable development, eco-efficiency, and green chemistry have dramatically influenced scientific research, addressing the development of new biomaterials derived from natural resources (e.g., from crops, vegetation, plants, animals, and bacteria) [1,2]. Natural polymers have been used extensively to design innovative devices suitable for different applications including wound dressings, drug delivery, vascular grafts, and tissue engineering [3,4]. They include chitosan [5,6,7], gelatin [8], alginate [4], starch [9], cellulose [4,10,11,12,13], and polyvinyl alcohol [12]. Among them, cellulose nanofibers (CNFs) are uniquely beneficial due to their high specific surface area, biodegradability, biocompatibility, and sustainability properties [14,15]. Furthermore, CNFs can easily be derived from cellulose sources (i.e., wood. and plant fibers) by using different techniques, including mechanical shearing or chemical and enzyme pretreatments combined with mechanical disintegration [14,16,17].

Recently, CNFs were proposed in combination with inorganic materials for the preparation of biocomposites with unique biological/biomechanical responses [16]. In particular, calcium phosphates have been recently proposed in combination with CNFs to design extracellular matrix analogues suitable for bone regeneration [18]. Traditionally, they have been embedded in biodegradable matrices with specific requirements, in terms of biocompatibility, timely controlled biodegradability, and biomechanical properties [19,20,21] to fabricate biomimetic scaffolds that can replace or regenerate portions of damaged or diseased hard tissues. They have generally been made of synthetic polymers such as aliphatic polyesters (polylactic acid (PLA), poly (glycolic acid) (PGA), poly (ε-caprolactone) (PCL), poly (dioxanone) (PDO), and poly (trimethylene carbonate) (PTMC) [22,23,24,25,26]. Recently, the discovery of new formulations involving natural polymers (i.e., proteins and/or polysaccharides) in combination with innovative processing techniques is leading to design composite materials with multiscale functionalities for bone healing/repair processes [27]. In particular, alginates are a mixture of two monosaccharides, i.e., guluronic and mannuronic acids, which can interact with divalent cations under normal physiological conditions to form cationic gels [28] and are promising materials to fabricate bioinspired platforms for efficaciously hosting cells and molecular signals [29]. In this view, a variety of experimental studies have been conducted to promote the mineralization of polysaccharide-like matrices via controlled calcium phosphate deposition [30,31,32], also suggesting innovative therapeutic solutions to support healing processes of hard [33] and soft [34] tissues.

From the processing point of view, recent studies have demonstrated that in situ precipitation strategies can be considered to be an alternative but effective route to manufacture CaP embedded composite matrices [35]. In these cases, polymer solutions containing phosphate (or calcium) ions can be dropped into calcium (or phosphate) solutions or, alternatively, polymer substrates immersed in calcium chloride and sodium dihydrogen phosphate anhydrous solutions with a Ca/P ratio of 1.67. By the reaction between calcium and phosphate ions, apatite-like phases with different structures and compositions may be confined into the bulk or onto the surface of the polymer matrices. Similar methods have been implemented to optimize the mineralization of collagen/nano HAp [12], alginate/nano HAp, and chitosan/nano HAp composites [36].

Taking into consideration the previous studies, here, we propose fabricating mineralized hybrid hydrogels formed by in situ precipitation of calcium phosphates in alginate/PVA matrices reinforced by the TEMPO-oxidized cellulose nano-fibrils and calcium ion interactions. In this case, phosphate ion rich solution is added to trigger the calcium phosphate mineralization process to produce alginate/PVA/CNF hybrid materials. The mineralized hybrid hydrogels processed at different pH conditions (pH 4 and 8) are systematically investigated in terms of morphology and microstructure of mineral deposits. Moreover, in vitro tests with HFB-4 and HSF skin cells are assessed to preliminary investigate cytotoxicity and cell viability in the presence of the proposed materials.

## 2. Materials and Methods

### 2.1. Materials

The bleached bagasse pulp was supplied from Quena Company of Paper Industry, Egypt. The hydrolyzed polyvinyl alcohol (PVA) (average Mw 30,000–70,000) and sodium alginate (SA) (Mw = 1.93 × 105 g/mol) were obtained from Sigma-Aldrich, USA. Sodium metaperiodite (NaIO_4_), NaBr, and 2,2,6,6-tetramethylpiperidine-1-oxyl (TEMPO) were purchased from Sigma-Aldrich (St. Louis, MA, USA). Additional chemicals needed for the different analytical methods were bought from Sigma-Aldrich (St. Louis, MA, USA). All chemical reagents were used with no further purification.

### 2.2. Preparation of TEMPO-Oxidized Cellulose Nanofibrils

The TEMPO-oxidized bleached bagasse pulp was made according to the method by Saito et al. (2007) [37]. The dispersion of bleached bagasse pulp (5 g) in distilled water (500 mL) was carried out using TEMPO (0.08 g, 0.5 mmol) and sodium bromide (0.8 g, 8 mmol). The next step was to add 50 mL of sodium hypochlorite solution, and then adjust the pH to 10. Afterward, the pH was adjusted to 7 and centrifuged at 7000 rpm. Further purification was achieved by repeating the addition of water, dispersion, and centrifugation. After 1 week of dialysis against deionized water, the product was filtered once again.

An electric conductivity titration method was used to measure the carboxylate content of the T-CNFs. This suspension was prepared by thoroughly mixing a dried sample with 0.01 M HCl (15 mL) and deionized water (20 mL), and vigorously stirring the mixture to obtain a well-dispersed suspension. Titrations were performed with sodium hydroxide (NaOH) solution of 0.01 M. The carboxylate content of the T-CNFs was determined by sudden changes in conductivity.

Equation (1) was used to calculate carboxylate content, C (mmol/g) as follows:C = ((V_1_ − V_0_) × C_NaOH_)/m(1)
where V_1_ and V_0_ are the volumes of standard NaOH solution before and after titration, respectively; C_NaOH_ is the concentration of standard NaOH solution; and m is the weight of the dried sample. The T-CNFs were prepared using a Masuko grinder (Honcho, Japan) for the mechanical defibrillation treatment using five passes and a constant frequency of 1500 rpm.

### 2.3. Preparation of Alginate/PVA/Calcium Phosphate Hybrid Material

To prepare the mineralized alginate/PVA and CNF hybrids, 2.5 g of polyvinyl alcohol was dissolved in 50 mL of water. Thereafter, 10% T-CNFs (10% w) were added, in order to form a 50:50 CNF/PVA solution. Moreover, 0.1 M HCl was vigorously mixed at 40 °C for 4 h to adjust the pHs of the solutions down to 4. Lastly, 5 g of sodium alginate was gradually added until a homogeneous paste was formed under stirring at room temperature. The paste was immersed in CaCl_2_ (0.1 M) and vigorously mixed for a while. The pH was adjusted to 4 and 8 with NaOH 0.1 M. Then, the forming hybrid hydrogel solution was gently stirred for 3 days at room temperature with a phosphate ion rich solution until a 0.24 M calcium concentration and a 0.12 M phosphate concentration were reached. Lastly, the composite mixture was stirred gently for 7 days at room temperature, and then thoroughly washed three times in deionized water for further characterization.

### 2.4. Morphological and Microstructural Characterizations

A scanning electron microscope (FEI-Quanta 200 FEG -ESEM, Eindhoven, The Netherlands) was used to determine the surface morphology of the mineralized hybrid materials at different pHs. Meanwhile, the morphology of nanometric phases was investigated via Transmission Electron microscope (TEM) with a JEOL (JEM-2100—Tokyo, Japan) electron microscopy setup at 100 k* magnification and acceleration voltage of 120 kV. The infrared spectra (FT-IR) were obtained using KBr discs (Perkin Elmer, Waltham, MA, USA), in the range 4000–500 cm^−1^ with a 4 cm^−1^ resolution and an accumulation of 16 scans for each analysis. A thermogravimetric analysis was performed on a TGA thermogravimetric analyzer (PerkinElmer, Waltham, MA, USA) under air from 25 to 800 °C with a heating rate of 10 °C/min. The X-ray diffraction (XRD) patterns were recorded with an Empyrean Powder Diffractometer, The Netherlands (Cu Kα, 0.154 nm) between 5 and 70° 2θ with a step size of 0.01°/s.

### 2.5. Biocompatibility

In vitro studies were performed by using 10^4^ cells/well HFB-4 (fibroblast normal cells derived from human skin tissue) and HSF (somatic normal cells derived from human skin tissue) lines obtained from the American Type Culture Collection (ATCC, Manassas, VA, USA) via Egyptian VACSERA. Cell viability was assayed using the colorimetric MTT procedure, as demonstrated by Mosmann (1983) and El-Fakharany et al. (2020). In brief, both normal cell lines were seeded in two sterile 96-well flat bottom microplates at a concentration of 1.0 × 10^4^/well and cultured in supplemented DMEM media (Lonza, Milan, Italy) with 10% fetal bovine serum (FBS) at 37 °C, in a 5% CO_2_ incubator. After overnight incubation, different doses from two crosslinked alginate/PVA/calcium phosphate (at pH 4 and 8) hybrid materials at concentrations of 0.125, 0.25, 0.5, 1.0, 1.5, and 2.0 mg/mL were added separately to each cell line, in triplicate, and incubated for 1, 3, and 7 days in a 5% CO_2_ incubator at 37 °C. After incubation times, the cells were washed 3 times with fresh media to remove dead cells and debris, then, 200 μL of MTT (3-(4,5-dimethylthiazol-2-yl)-2,5 diphenyltetrazolium bromide) solution at a concentration of 0.5 mg/mL was added to each well. After incubation for 2–3 h in a 5% CO_2_ incubator at 37 °C, the MTT solution was removed and substituted with 200 μL/well DMSO to dissolve the formed formazan crystals. All tested cells were measured at 570 nm using a microplate ELISA reader (BMG LabTech, Munich, Germany) and the relative cell viability (%) as compared with control wells containing cells without treatment was calculated using the following formula: (X) test/(Y) control × 100%. In addition, the sensitivity of crosslinked alginate/PVA/calcium phosphate (at pH 4 and 8) hybrid materials at concentrations of 0.5 and 1.0 mg/mL against the morphology of both HFB-4 and HSF cells was estimated by visualization under a phase contrast microscopy (Olympus, Hamburg, Germany) and compared to untreated control cells [38,39].

### 2.6. Statistical Analysis

All experimental trials were tested in triplicate (*n* = 3) and all data were presented as mean ± SEM. Statistical analysis significance was evaluated by the multiple comparisons test of the one-way analysis of variance (ANOVA) using graph Pad Prism software 6.0, and differences were considered to be statistically significant at *p*-values < 0.05.

## 3. Results and Discussion

Wound and bone fracture healing involve natural repair processes due to traumatic events. The use of biomaterials may offer an extraordinary opportunity to support native capabilities of human bone and skin to self-regeneration and repair [40]. Recently, a great deal of attention has been devoted to materials from sustainable resources, motivated by increasing requests to identify new bio-recognized alternatives to traditional biomaterials from synthetic and natural sources [41]. Among them, cellulose fibers from bagasse residues have been recently demonstrated to be an excellent material that supports cell interactions. Because of their high cellulose content (around 70%) and their wide availability as waste materials, they can be considered to be an excellent cellulosic source for nanocellulose production. Their pretreatment via TEMPO oxidation may enable them to convert the primary hydroxyl groups in C6 to carboxylates for the preparation of nanofibrils (T-CNFs) with improved biocompatibility. In this work, TEMPO-oxidized celluloses containing more than 1 mmol/g sodium carboxylates were suspended in water and subjected to gentle mechanical disintegration, until transparent and highly viscous gels were produced (Figure 1A). The morphology of the T-CNFs was investigated using transmission electron microscope (TEM). The average diameter of nanofibers isolated in Figure 1B are homogeneous, ranging from 6 to 15 nm, while lengths fall into a range of a few micrometers. Figure 1C shows an X-ray diffraction pattern produced by a T-CNF compound which shows peaks at 16° and 22.4°, which correspond to (101) and (002), thus, confirming a carboxylate group arrangement on the surface. In addition, a peak was observed at 34.2° more or less parallel to the (040) plane. The FTIR spectrum of cellulose confirmed TEMPO oxidation, since a new band corresponding to stretching of carbonyl groups was detected at 1745 cm^−1^ (Figure 2D).

Once optimized the CNF formation, the mineralized hybrid hydrogels were prepared by combining CNFs with crosslinked alginate/PVA gels followed by in situ precipitation of synthesized *HA*-like crystals by controlling the interaction with carboxymethyl cellulose-g-polymethacrylic acid groups. As a function of the pH conditions, they were investigated in terms of morphological, physical, chemical, and thermal properties.

Figure 2 shows the SEM analysis of the T-CNF/alginate/PVA and T-CNF/alginate/PVA/calcium phosphate composites prepared via calcium phosphate precipitation at different pHs. The T-CNF/alginate/PVA showed a porous surface with a high degree of homogeneity, which confirmed good cohesion of alginate, PVA, and T-CNFs. In particular, the precipitates of calcium phosphate at pH 4 (Figure 2B) exhibited plate-like structures with different spatial configurations, i.e., tangled or paralleled. Furthermore, the EDX results showed that the Ca/P ratio of the formed calcium phosphate was 1.2, close to DCPDH’s stoichiometric ratio, confirming the formation of brushite-like phases in the T-CNF/alginate/PVA hybrid materials. Contrariwise, at pH 8, the precipitated phases of calcium phosphate tended to form spherical aggregates with uniform size, strictly packed into the alginate/PVA matrix. Accordingly, the EDX analyses highlighted an increase in the Ca/P ratio at pH 8, and the formation of stoichiometric hydroxyapatite crystalline phases (Ca/P 1.76).

In order to better investigate the internal structure of calcium phosphates after the biomimetic treatment, investigations via transmission electron microscopy (TEM) were further assessed. It was observed that the mineralized hydroxyapatite nanocrystals formed at pH = 8, on the one hand, were characterized by a peculiar rod-like shape with a narrow size distribution falling in the nanometer size range (Figure 3B). This is ascribable to their active role as catalyzing agent of PVA and alginate interactions during calcium phosphate precipitation at slightly alkaline conditions. On the other hand, at pH = 4, the calcium phosphates tended to form thin platelets, as shown in Figure 3A.

In order to further investigate the microstructural properties of calcium phosphates dispersed in the matrix, FTIR analyses were assessed. As reported in Figure 4, characteristic polysaccharide peaks at 899, 1060, 1150 cm^−1^ were associated with the vibrational mode of C-O stretching and COC-bridge stretching from glycosidic bonds. Accordingly, absorption peaks corresponding to phosphate groups were detected in the spectra of the T-CNF/alginate/PVA hybrid materials. These peaks referring to between 1050:1060 cm^−1^ (P–O υ3 mode) round 560 cm^−1^ (P–O υ4 mode) and 899 cm^−1^ (P–O υ1 mode) provided a confirmation that the mineralization process had occurred.

Figure 5 shows the XRD spectra of the hybrid hydrogels before and after the precipitation of calcium phosphate materials at different pHs. Two characteristic peaks at 2θ = 10.3° and 22° were attributed to a semi-crystalline material structure of the polysaccharide chains, i.e., cellulose and alginate. Noteworthy, calcium phosphates formed at pH 4, showed characteristic diffraction peaks at 2θ = 11.6°, 20.9°, 22.2°, 26°, 29.0°, 30.5°,31.4° 34.1° 37.0°, 38.3°, 39.9° and 42.1°, typically assigned to dicalcium phosphate dehydrate (DCPDH) crystals. Otherwise, calcium phosphate formed at higher pH conditions (i.e., pH 8) presented diffraction patterns with characteristic signals at 2θ = 2θ = 25.8°, 32°, 38.3°, 42.2° and 44.6°, that are typically recognized for the hydroxyapatite.

In order to investigate the biocompatibility of mineralized hybrid hydrogels, in vitro tests were performed with HFB-4 and HSF cells. All the experiments were conducted in order to investigate the response of cells only in the presence of foreign materials, in agreement with the proposed healing applications (see experimental section). Figure 6 summarizes the response of HFB-4 and HSF cells in terms of cell viability in the presence of different hybrid hydrogels with different concentrations, after 1, 3, and 7 days in culture. The results indicated that no cytotoxic response against HFB-4 and HSF cells was induced by hybrid hydrogels with low concentrations, until 0.5 mg/mL, independently of the culture time (i.e., 1, 3, or 5 days). Only a slight cytotoxicity was recognized for a sample concentration equal to 1.0 mg/mL with an average 20% decrease in cell viability as compared with previous samples at 7 days. The highest cytotoxic response was measured in the case of sample concentration of 2.0 mg/mL after 7 days incubation, where only 10% cell viability was recorder as compared with the positive lowest concentrated samples (Figure 6). These trends were further confirmed by inverted phase contrast images via optical microscopy (Figure 7).

All the results (Figure 6 and Figure 7) suggested that cell viability is sensitive to mineral phase changes ascribable to the pH conditions used during in situ precipitation.

In addition, we have demonstrated that a switch from slightly acidic to slightly alkaline conditions triggered the precipitation of dicalcium phosphate dehydrate (DCPDH) and hydroxyapatite, respectively. Taking this into consideration, the biological results are not surprising. Indeed, according to several studies reported in the literature, it is well known that ion reactivity of calcium phosphates, as well as their attitude to release different amounts of ions (i.e., Ca^2+^, Mg^2+^, and HPO_4_^2−^), can be closely correlated with the local and temporal pH changes occurring in in vitro culture, thus, influencing cell behavior [42]. In this case, dicalcium phosphate dehydrates tend to convert in the hydroxyapatite form in in vitro culture, thus, releasing a pattern of selected ions that negatively influence the viability of cells, until showing a cytotoxic response for higher concentrations, in agreement with previous experimental evidence [43]. Moreover, we suggest that hybrid hydrogels may also locally influence environmental calcium levels, by releasing calcium in the acidic environment with relevant effects, as a function of pH, in bone or wound healing mechanisms as suggested in previous similar works [44,45].

## 4. Conclusions

In this work, CNFs were applied as crosslinkers to reinforce alginate/PVA hydrogels. Calcium ions were further used to improve the bioactivity by a biomimetic process of in situ mineralization and to corroborate the gelation mechanism of the alginate/PVA network. The mineralized hybrid hydrogels were processed at different pH conditions (pH 4 and 8) and optimized in terms of morphology and microstructure of mineral deposits. Whilst plate-like structures were recognized at pH 4, uniformly dispersed clusters of round-like crystal aggregations were observed at pH 8; the chemical and physical analyses identified, respectively, di-calcium phosphate dehydrate and hydroxyapatite phases in the polymer matrices. In vitro studies confirmed a low cytotoxicity of all the hybrid hydrogels at low concentrations, but highlighted a significant decay of cell viability in the presence of highly concentrated gels including di-calcium phosphate dehydrate crystals that tended to catalyze a massive ion release and, consequently, a negative response of cells in vitro. On the basis of these preliminary results, PVA/Alg/CNF gels endowed with HAp phases are identified as the more stable formulation to be used in vivo to support healing processes at the interface with bone and/or skin wounds.

## Figures and Tables

**Figure 1 molecules-27-00697-f001:**
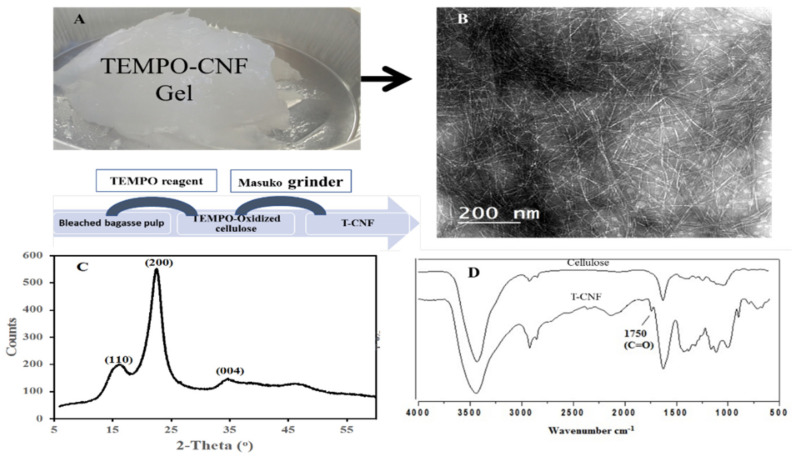
Preparation of TEMPO-oxidized cellulose nanofibrils dispersed in water by mechanical disintegration in water: (**A**) Photograph of the highly viscous gel; (**B**) TEM; (**C**) XRD; (**D**) FTIR of cellulose and T-CNFs.

**Figure 2 molecules-27-00697-f002:**
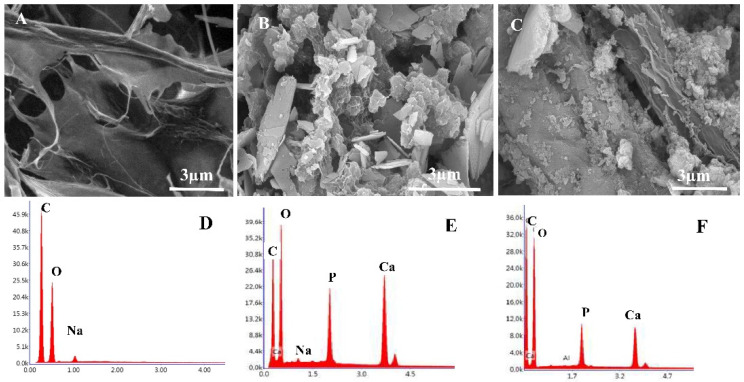
SEM images and EDX analysis of the T-CNF/alginate/PVA hydrogel before (**A**,**D**) and after calcium phosphate mineralization at pH 4 (**B**,**E**) and at pH 8 (**C**,**F**).

**Figure 3 molecules-27-00697-f003:**
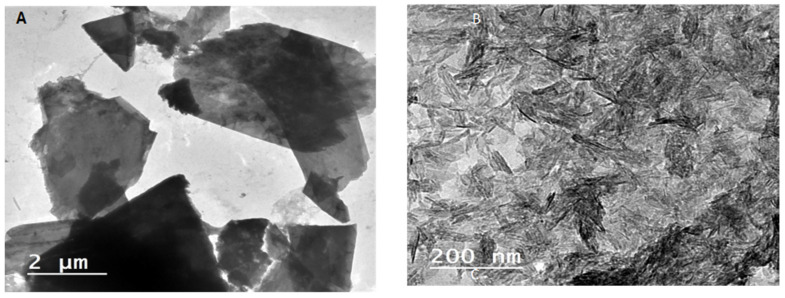
TEM images of T-CNF/alginate/PVA/calcium phosphate mineralization: (**A**) at pH 4; (**B**) at pH 8.

**Figure 4 molecules-27-00697-f004:**
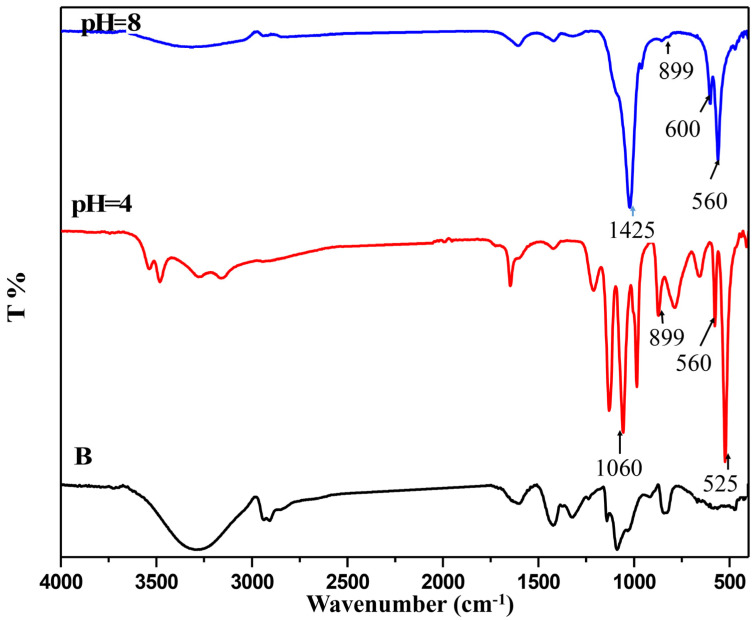
FTIR spectra of (B) Crosslinked alginate/PVA and crosslinked alginate/PVA/calcium phosphate hybrid materials at pH 4 and 8.

**Figure 5 molecules-27-00697-f005:**
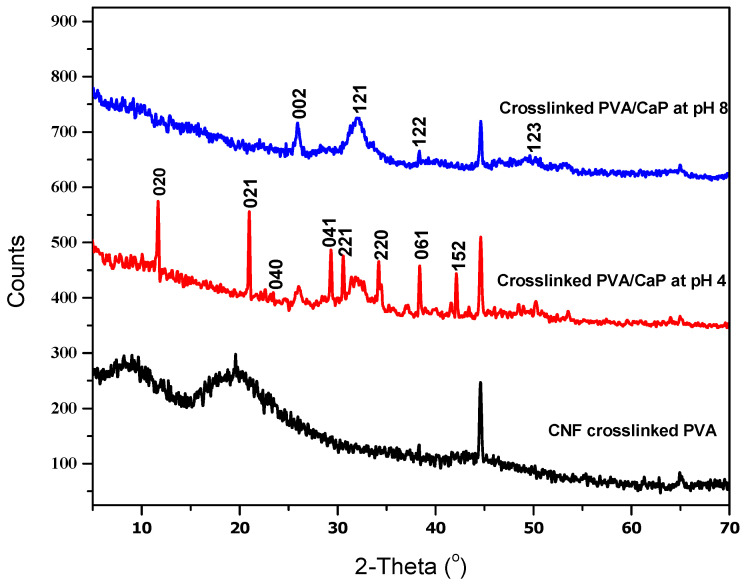
XRD patterns of PVA/T-CNF, and PVA/T-CNF/calcium phosphate hybrid materials formed at pH 4 and 8.

**Figure 6 molecules-27-00697-f006:**
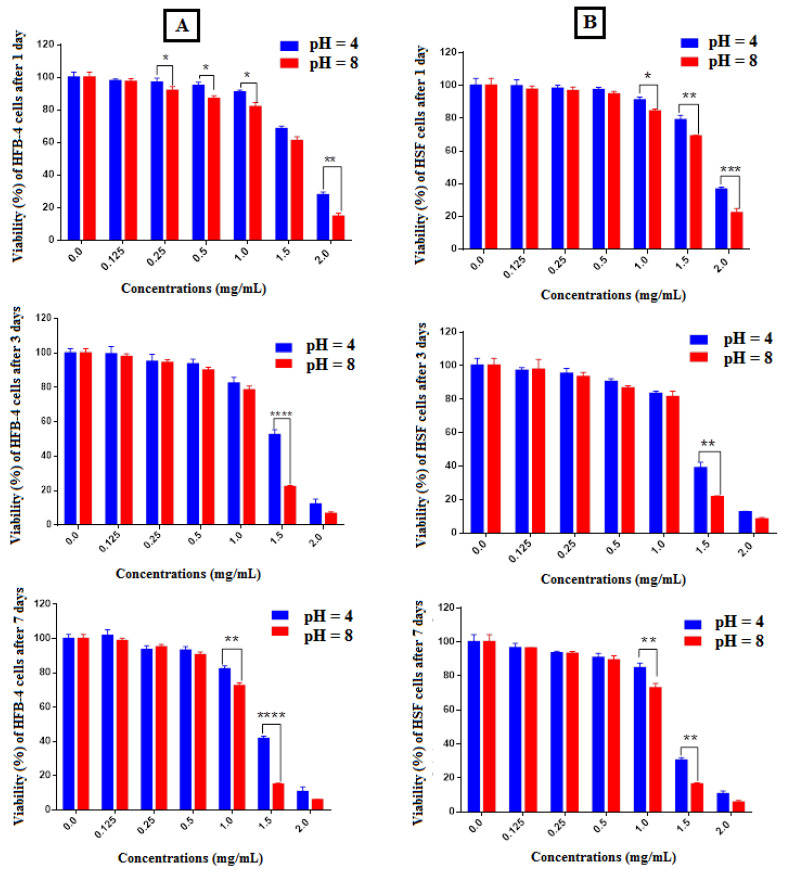
Effect of crosslinked alginate/PVA/calcium phosphate hybrid materials on the viability of HFB-4 (**A**) and HSF (**B**) cells. The normal cell lines were both incubated with tested samples at various doses (0.125–2.0 mg/mL) for 1, 3, and 7 days. The cell viability test was assessed by using MTT assay. All values represent the average values from three experiments and expressed as mean ± SEM. (* *p* ≤ 0.025 vs. ** *p* ≤ 0.01 vs. *** *p* ≤ 0.001 vs. **** *p* ≤ 0.0001 vs. crosslinked alginate/PVA/calcium phosphate hybrid materials, mean ± SEM, *n* = 3).

**Figure 7 molecules-27-00697-f007:**
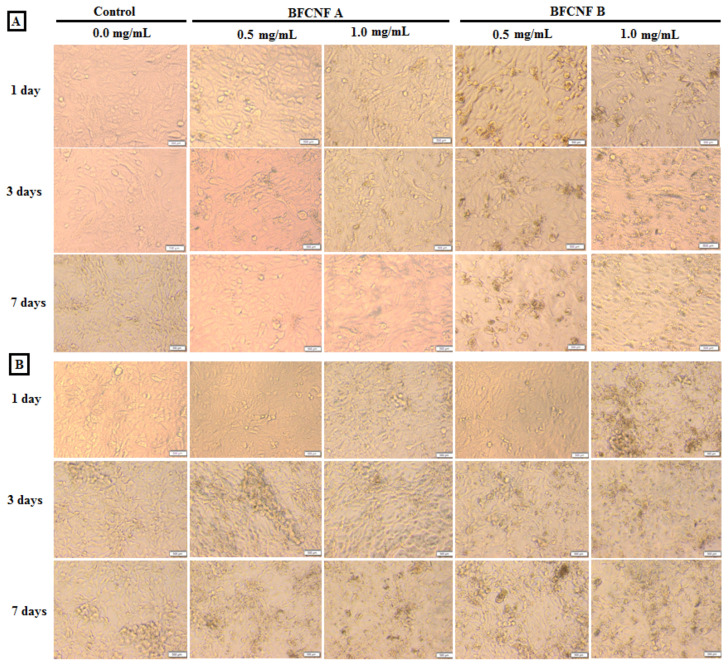
(**A**) and HSF (**B**) cells as visualized under inverted phase contrast microscope. The prepared samples were added at different concentrations of 0.0 μg/mL (control), 0.5 mg/mL, and 1.0 mg/mL to both tested cells, and incubated for 1, 3, and 7 days.

## Data Availability

Not applicable.

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
