# Peer review of "Mineralized Polyvinyl Alcohol/Sodium Alginate Hydrogels Incorporating Cellulose Nanofibrils for Bone and Wound Healing"

_molecules, 2022, doi:10.3390/molecules27030697_

Round 1

Reviewer 1 Report

The article Mineralized Hybrid Hydrogels Embedding Cellulose Nano- 2 fibrils for Biomedical Applications is an interesting article describing preparation of stable mineralized hydrogel of TEMPO-oxidized cellulose nanofibrils embedded to alginate and polyvinyl alcohol. In addition calcium chloride reaction was optimized to trigger the polymer chains cross-14 linking reaction and mutually promoting the in-situ mineralization of calcium phosphates.

Nevertheless, there are parts of the text that need either clarification of require correction, therefore I suggest a major revision of the article. If the authors are willing to address the following issues or comment on the below stated suggestions accordingly, I would reconsider the publication of this article in the Moleculs.

GENRAL REMARKS

  1. The authors should more clearly state the novelty of their approach in comparison with the similar research.

CONCRETE REMARKS

Introduction

  1. Page 1, line 33: before [3,4] please remove one space
  2. Page 1, line 36: since human don´t have the enzymes for cellulose degradation, is it right, to use the term biodegradable?? Please add the explanation what happens with CNF in the body!
  3. Page 2, line 73: after Hap please remove one space

Material and methods

Everywhere the city and country of producers should be added: Quena, Sigma Aldrich, FEI-Quanta, TEM, FT-IR, TGA, microplate ELISA reader…

  1. Page 3, line 97: the sentence should be corrected
  2. Why you decided for pH 4 and 8? You should discuss that at least in results!

Results and discussion

  1. Please mark on IR spectrum where a new peak is formed (add the curve before TEMPO oxidation and then on the graph clearly mark in the text mentioned peak).
  2. Page 5, line 202: a space to much
  3. Page 7, line 232: a space to much
  4. Page 8, lines 261-272: you are writing about influence of pH on cell viability where the differences are significant from 1 mg/ml on… Regarding not so good results of viability, the Live Dead test directly on newly formed material should be performed!
  5. Based on what you decided for HFB-4 and HSF cells? Namely your conclusion talks about the regeneration of hard tissues like bones!!

Author Response

The article Mineralized Hybrid Hydrogels Embedding Cellulose Nano- 2 fibrils for Biomedical Applications is an interesting article describing preparation of stable mineralized hydrogel of TEMPO-oxidized cellulose nanofibrils embedded to alginate and polyvinyl alcohol. In addition, calcium chloride reaction was optimized to trigger the polymer chains cross-14 linking reaction and mutually promoting the in-situ mineralization of calcium phosphates.

Nevertheless, there are parts of the text that need either clarification of require correction, therefore I suggest a major revision of the article. If the authors are willing to address the following issues or comment on the below stated suggestions accordingly, I would reconsider the publication of this article in the Molecules.

The authors should more clearly state the novelty of their approach in comparison with the similar research.

- The current article contains a high degree of novelty. It is the first time to use CNF as crosslinker to prepare Alg/PVA hydrogel. Moreover, the biocompatibility investigation of CNF/Alg/PVP/calcium phosphate hybrids was reported in the current article. These results are promising for further developing of these hybrids in biomaterials. These points were illustrated in the manuscript.

Introduction

Page 1, line 33: before [3,4] please remove one space

- The space was removed

Page 1, line 36: since human don´t have the enzymes for cellulose degradation, is it right, to use the term biodegradable?? Please add the explanation what happens with CNF in the body!

- Thank you for your comment. In biomedical applications, cellulose-based materials are widely used due to their recognized biocompatibility, for in vitro and in vivo applications. Biodegradability relates to the material’s ability to degrade by microorganisms, while bioresorbability refers to its ability to be digested or metabolized in vivo when implanted. One long-term study by Martson found that cellulose sponge implants degrade slowly in rats’ subcutaneous tissue. Taking into consideration that short times used to investigate the degradation (60 weeks) and current limits of knowledge about the mechanism about in vivo resorption, cellulose-based implants should be considered bio-durable. However, it is possible to produce resorbable cellulose-based devices by chemically modifying cellulosics with bioresorbable molecules as reported in the following work [Martson M., Viljanto J., Hurme T., Laippala P., Saukko P. Is cellulose sponge degradable or stable as implantation material? An in vivo subcutaneous study in the rat. Biomaterials. 1999;20(21):1989–1995. doi: 10.1016/S0142-9612(99)00094-0]. This reference has been included in the introduction section.

Page 2, line 73: after Hap please remove one space

- The space was removed

Material and methods

Everywhere the city and country of producers should be added: Quena, Sigma Aldrich, FEI-Quanta, TEM, FT-IR, TGA, microplate ELISA reader…

- The manuscript has been amended accordingly

Page 3, line 97: the sentence should be corrected

Why you decided for pH 4 and 8? You should discuss that at least in results!

- Thank you for your comment.  We applied various pH to obtain various crystalline forms of calcium phosphate, especially hydroxyapatite and brushite. The cytotoxicity of the formed hybrid was also investigated.

Results and discussion: Please mark on IR spectrum where a new peak is formed (add the curve before TEMPO oxidation and then on the graph clearly mark in the text mentioned peak).

- The IR of cellulose was added, and the main peak was marked.

Page 5, line 202: a space to much

- The space was removed

 Page 7, line 232: a space to much

- The space was removed

Page 8, lines 261-272: you are writing about influence of pH on cell viability where the differences are significant from 1 mg/ml on…

Regarding not so good results of viability, the Live Dead test directly on newly formed material should be performed!

- As for your comment, we would underline that MTT method is a rapid reliable method and used for calculating both cell viability and cytotoxicity.

Based on what you decided for HFB-4 and HSF cells? Namely your conclusion talks about the regeneration of hard tissues like bones!!

- Thank you for your comment. Human fibroblast (HFB-4) and derived skin cells (HSFs) are cell lines typically used for testing the toxicity in wound healing applications. However, they can be also successfully used to preliminary evaluate cytotoxic response in the presence of bioactive compounds such as calcium phosphates [DOI: 10.1007/s10904-018-0864-1; doi.org/10.1016/j.jascer.2014.05.003].  In the next future, we will provide an accurate investigation of in vitro response of proposed materials buy using osteoblasts (now still available in our lab) for a more accurate validation for bone applications.

Reviewer 2 Report

Reviewer : 1

  1. Introduction

The introduction section is long with  many references based on the literature survey conducted by the authors. This is very good. However, this lacks in proper presentation of literature survey, which should have been systematic whereby existing scientific gaps should have been brought out. This should have given justification for the present study, which should be followed by the objectives of this study.

  1. Mineralized hybrid hydrogels processed at different pH conditions (pH 4 and 8) were systematically investigated in terms of morphology and microstructure of mineral deposits. Moreover, in vitro tests with HFB-4 and HSF skin cells were assessed to validate the biocompatibility of proposed materials.

why the choice of parameters at pH 4 and 8 for this study. Additional some related references about these pH parameters?

  1. Result and discussion :

this Section is Results & Discussion and hence only results should be mentioned and then it should be discussed preferably comparing it with earlier reported similar results by other researchers

Author Response

Introduction: The introduction section is long with many references based on the literature survey conducted by the authors. This is very good. However, this lacks in proper presentation of literature survey, which should have been systematic whereby existing scientific gaps should have been brought out. This should have given justification for the present study, which should be followed by the objectives of this study.

Mineralized hybrid hydrogels processed at different pH conditions (pH 4 and 8) were systematically investigated in terms of morphology and microstructure of mineral deposits. Moreover, in vitro tests with HFB-4 and HSF skin cells were assessed to validate the biocompatibility of proposed materials.

why the choice of parameters at pH 4 and 8 for this study. Additional some related references about these pH parameters?

Thank you for your comment. We applied various pH to obtain various crystalline forms of calcium phosphate, especially hydroxyapatite and brushite. The cytotoxicity of the formed hybrid was also investigated. Additional references were added to prove the pH Parameter. Calcium phosphate mineralization controlled by carboxymethyl cellulose-g-polymethacrylic acid.

Result and discussion: this Section is Results & Discussion and hence only results should be mentioned and then it should be discussed preferably comparing it with earlier reported similar results by other researchers.

Thank you for your comment. The results and discussion section has been improved in agreement with the reviewer suggestion. Some introductory sentences have been included in the first part of the section to contextualize the objective as follows “Wound and bone fracture healing involve natural repair processes due to traumatic events. The use of biomaterials may offer an extraordinary opportunity to support native capabilities of human bone and skin to self-regeneration and repair [42]. In the last decade, a great deal of attention has been devoted to the investigation of biological interaction of materials from sustainable resources, moved by the increasing request to identify new bio-recognized alternatives to traditional biomaterials from synthetic and natural sources [43]. Among them, cellulose fibres from bagasse residues have been recently demonstrated to be an excellent material that support cell interactions”. Then, the discussion section has been improved by adding a sentence to better underline the role of calcium phosphates in proposed hybrid hydrogels for the specific applications also including 2 new references, as follows “Moreover, we suggest that hybrid hydrogels may also locally influence environmental calcium levels, by releasing calcium in the acidic environment with relevant effects - as a function of pH – on bone regeneration or wound healing mechanisms as suggested in previous similar works [44, 45]”

Reviewer 3 Report

Dear authors! The manuscript "Mineralized Hybrid Hydrogels Embedding Cellulose Nano-2 fibrils for Biomedical Applications" is a very well-planned paper with clear results that should be published after some revision. Additionally to my comments below, I would like to recommend a better description of the rationale for material application. Also, more discussion is required to make this manuscript more scientifically sound. 

  1. The title of paper is confused and required modification.
  2. From the abstract it is not clear the aim of current hydrogel development. The general phrase “is still a relevant challenge in biomedical field” is not enough.
  3. In abstract, the phrase “for different applications in hard tissue repair” is general and not clear.
  4. In introduction: “hard tissues like bone” – what other hard tissue we can expect in human body? If you propose some material for bone regeneration. It should be clearly demonstrated. In my opinion, authors need to remove phrase “hard tissue” to “bone”.
  5. Methodology section, “Biocompatibility”. You need to share a source of cells.
  6. At the beginning of this section you told about 104 cells per well. Later – about 103 cells per well. Please, clarify.
  7. “microplate ELISA reader” – please, clarify, type, producer, etc.
  8. What statistical calculation you used? Please, clarify.
  9. Figure 2 – scale bare require on SEM image.
  10. Biocompatibility results. “with an average 20% decrease in cell viability” – is it statistically significant?
  11. Different pH did not describe in biocompatibility methodology. It should confuse readers – it is not clear is it pH of media or samples. Please, clarify and put clear explanation to methodology section.
  12. “All these results suggest that cell viability is sensitive to the pH conditions used during the in situ precipitation” – I do not see this from the results. There is no any difference in cell viability depending on pH.
  13. The figure 7 is low quality. You can provide some results as supplement. More over, this figure is not informative.

Author Response

Dear authors! The manuscript “Mineralized Hybrid Hydrogels Embedding Cellulose Nano-2 fibrils for Biomedical Applications” is a very well-planned paper with clear results that should be published after some revision. Additionally to my comments below, I would like to recommend a better description of the rationale for material application. Also, more discussion is required to make this manuscript more scientifically sound.

The title of paper is confused and required modification.

Thank you for your comment. The title has been modified for better clarity as follows: “Mineralized Polyvinyl alcohol/Sodium alginate Hydrogels incorporating Cellulose Nanofibrils for Biomedical Applications”

From the abstract it is not clear the aim of current hydrogel development. The general phrase “is still a relevant challenge in biomedical field” is not enough.

Thank you for your comment. The sentence has been rewritten appropriately as follows “Bio sustainable hydrogels including  tunable morphological and/or chemical cues currently offer a valid strategy of designing innovative systems to enhance healing/regeneration processes of damaged tissue areas.

In abstract, the phrase “for different applications in hard tissue repair” is general and not clear.

Thank you for your comment. The sentence has been amended as follows “for different applications in wound and bone healing”.

In introduction: “hard tissues like bone” – what other hard tissue we can expect in human body? If you propose some material for bone regeneration. It should be clearly demonstrated. In my opinion, authors need to remove phrase “hard tissue” to “bone”.

Thank you for your comment. The text has been corrected according with the reviewer suggestions.

Methodology section, “Biocompatibility”. You need to share a source of cells.

Thank you for the comment. We obtained the cell lines from ATCC. This has been included in the materials and methods section.

At the beginning of this section you told about 104 cells per well. Later – about 103 cells per well. Please, clarify.

Thank you for your comment, The text has been amended and cell number corrected.

“microplate ELISA reader” – please, clarify, type, producer, etc.

Thank you for your comment, The text has been amended accordingly

What statistical calculation you used? Please, clarify.

Thank you for your comment.  Section 2.5 has been added to clarify this point.

Figure 2 – scale bare require on SEM image.

The scale bare on SEM image was added

Biocompatibility results. “with an average 20% decrease in cell viability” – is it statistically significant?

Thank you for the comment.  Significance data are now included into the manuscript (* P≤ 0.025 vs. ** P≤ 0.01).

Different pH did not describe in biocompatibility methodology. It should confuse readers – it is not clear is it pH of media or samples. Please, clarify and put clear explanation to methodology section.

Thank you for your comment, The text has been amended accordingly.

“All these results suggest that cell viability is sensitive to the pH conditions used during the in situ precipitation” – I do not see this from the results. There is no any difference in cell viability depending on pH.

Thank you for this question, pH changes refers to the conditions of mineral phase deposition. This has been clarified in the results and discussion section as follows “All these results suggest that cell viability is sensitive to mineral phase changes as-cribable to the pH conditions used during the in situ precipitation. As reported by data, there is a significant difference between two pHs at high concentrations of 1.5 and 2.0 mg/ml.

The figure 7 is low quality. You can provide some results as supplement. Moreover, this figure is not informative.

The quality of the figure was improved.

Round 2

Reviewer 1 Report

I suggested to do an additional live/dead test directly on the material, where you can show the influence of the material on the cells directly. 

Also, I suggested to perform the tests with the cells you are talking about (osteoblasts) or change the introduction/conclusion to apply to all the tests you perform. 

You have ignored both of these suggestions. If the other reviewers agree with the publication, I do not require you to perform additional tests, but I think it is necessary to change the diction of the proposed applicability according to the tests used in the research.

Author Response

Thank you for your comments. As for your first question, in our work, we have proposed to investigate the response of cells in proximity of hybrid hydrogels (not directly seeding cells onto the gels) in order to better reproduce the typical healing conditions that occur in vivo in the presence of host biomaterials. For this purpose, it is included a sentence in the results and discuss section to better clarify this aspect as follows: “All the experiments were conduit indirectly, as described in the experimental section, in order to investigate the response of cells only in the presence of foreign materials, in agreement with the proposed healing applications”. As for your second question, I agree with the reviewer about the use of osteoblasts to validate more appropriately the proposed materials for bone. However, it is also remarked that the use of somatic cells like HSF has been adopted in the past, only in preliminary way, to validate the biocompatibility of different biomaterials for bone and for applications in bone healing [Front. Endocrinol., 19 June 2020 https://doi.org/10.3389/fendo.2020.00394][Experimental & Molecular Medicine volume 51, pages1–8 (2019).

In agreement with the reviewer suggestion, in the conclusion section, the last sentence has been amended as follows: ”Starting from these preliminary results, PVA/Alg/CNF gels endowed with Hap phases are identified as the more stable formulation to be used in vivo to support healing processes at the interface with bone and/or skin wounds”.

Reviewer 3 Report

After revision paper is ready for publication.

Author Response

Thank you for your comment. Results and discussion section has been revised to improve the results description, in agreement with the reviewer siuggestion. all text amendments have been green marked